# Picking Winners: Identifying Features of High-Performing Special Purpose Acquisition Companies (SPACs) with Machine Learning

**Caleb J. Williams**

Data Science Division, Quilty Analytics, 33 6th St. S, Suite 204, St. Petersburg, FL 33701, USA;
cwilliams16@chicagobooth.edu

**Abstract:** Special Purpose Acquisition Companies (SPACs) are publicly listed "blank check" firms with a sole purpose: to merge with a private company and take it public. Selecting a target to take public via SPACs is a complex affair led by SPAC sponsors who seek to deliver investor value by effectively "picking winners" from the private sector. A key question for all sponsors is what they should be searching for. This paper aims to identify the characteristics of SPACs and their target companies that are relevant to market performance at sponsor lock-up windows. To achieve this goal, the study breaks market performance into a binary classification problem and uses a machine learning approach comprised of decision trees, logistic regression, and LASSO regression to identify features that exhibit a distinct relationship with market performance. The obtained results demonstrate that corporate or private equity backing in target firms greatly improves the odds of market outperformance one-year post-merger. This finding is novel in indicating that characteristics of target firms may also be deterministic of SPAC performance, in addition to SPACs, transaction, and the market features identified in the prior literature. It further suggests that a viable sponsor strategy could be constructed for generating outsized market returns at share lock-up windows by simply "following the money" and choosing target firms with prior involvement from corporate or private equity investors.

**Keywords:** Special Purpose Acquisition Company (SPAC); private equity; venture capital; Initial Public Offering (IPO); feature engineering; machine learning; artificial intelligence; classification and regression tree; logistic regression; LASSO regression; out-of-sample performance; econometrics; predictive analytics

## 1. Introduction

Special Purpose Acquisition Companies (SPACs) are publicly listed "blank check" firms with a sole purpose: to merge with another company and take it public. Wall Street's hottest deal trend has been rising in popularity for years, culminating in 2020, when SPACs outpaced non-SPAC IPOs for the first time to become the most popular type of new listing (MacArthur and Dessard 2020).

SPACs begin their lifecycle with "sponsors"—often former industry/banking executives or celebrities—forming a blank-check company that completes a traditional IPO to become a publicly traded entity (Gahng et al. 2021). Sponsors cover the initial underwriting costs and, in exchange, are compensated with equity and warrants that are commonly tied to 6- or 12-month lock-up agreements beginning once the merger has been executed (Carpenter et al. 2022). This unusual arrangement incentivizes sponsors to expend a costly effort to obtain information about firm characteristics and choose a high-quality firm for acquisition if they are to maximize their returns (Chatterjee et al. 2014; Feng et al. 2023).

For sponsors, the primary challenge is determining what characteristics they should be looking for in a merger target. This area has received little attention in the literature, as most prior performance determinant hypotheses have focused on the features of the

SPAC or the market, rather than the target firm (Dimitrova 2017). Concerning the relevant SPAC performance determinant research that does exist, much of it was published before the exponential rise in SPACs seen over the past four years (2018–2022), and the findings should be re-examined with the newly available data.

This paper's main contribution is investigating the characteristics of SPACs and target companies from the perspective of a SPAC sponsor and assessing how these features relate to market performance at the relevant sponsor lock-up windows. Ultimately, the research goal is to identify the relevant criteria for sponsors to consider during their SPAC search process. The work builds on prior performance determinant studies that have identified acquisition near the end of the pre-defined search timeframe, sponsor involvement in firm governance, and market volatility as critical features impacting SPAC performance one year after the transaction closed.

Following the above, this paper aims to give a precise answer to the following research question (RQ): what features of SPACs or their target companies should sponsors select to maximize market performance at sponsor lock-up windows?

To answer the RQ, the study uses a multi-phase machine learning approach: first, a binary classifier is constructed to indicate whether the SPAC under- or overperformed the market during its first year of trading post-de-SPAC. Next, the approach compares the feature selection results from decision tree and logistic regression models to identify potentially relevant features to the algorithm's predicted accuracy. Finally, a LASSO regression model is developed to isolate the critical features and identify those variables that are suitable for traditional testing as SPAC performance determinants. The use of basic, but fundamental machine learning models in this study is intended to enable a robust assessment of which features are potentially influential without the risk of overfitting on such a small dataset (n = 130).

Following the introduction, this paper is organized as follows: a concise literature review is presented in Section 2, describing prior research on both Special Purpose Acquisition Companies and using machine learning for predicting the performance of market securities. The materials and methods used for this analysis follow in Section 3, and an overview of the sample data, trends, and variable characterization is presented in Section 4. The results of the machine learning models and associated discussion are shown in Sections 5 and 6. Finally, Section 7 provides conclusions and implications of the effort for future researchers and practitioners.

## 2. Literature Review

### 2.1. Special Purpose Acquisition Companies (SPACs) and the Role of SPAC Sponsors

Early work on the topic of Special Purpose Acquisition Companies (SPACs) focused on the role of the sponsor incentive structure and characteristics of desirable management teams in maximizing market performance (Jog and Sun 2007; Berger 2008). The field was further expanded with research detailing the advantages of SPACs vs. traditional IPOs (Jenkinson and Sousa 2011; Floros and Sapp 2011). Near the same time, Boyer and Baigent began to consider SPAC performance, using regression models to estimate the impact of SPAC variables on first-day and one-year returns (Boyer and Baigent 2008). Their work demonstrated that the warrant, unit price/offer price, and time allowed to find an acquisition target all had statistically significant impacts on first-day returns; although, they did not perform this analysis on one-year returns. This is a key oversight in the current literature, as most present-day sponsors are subject to 6- or 12-month share lock-ups.

Beyond its performance determinant findings, Boyer and Baigent's work additionally provided a foundation for the exploration of underpricing as a proxy for the market's assessment of the management team (Murray 2014). During this period, the literature began to further explore the role of SPAC sponsors, demonstrating that SPAC compensation follows a similar model to that of private equity funds and suggesting that SPACs are a form of private equity (Rodrigues and Stegemoller 2011; Davidoff 2008). Since Murray, there have been few papers exploring the determinants of SPAC performance. Cumming

and their team evaluated the probability of various company/transaction characteristics in contributing to affirmative shareholders, finding, notably, that the presence of hedge and merger arbitrage funds led to fewer successful affirmative merger votes, but made no assessments of those characteristics in relation to performance (Cumming et al. 2014). Kolb and Tykvová looked extensively at market/firm determinants of SPACs, but their research was primarily focused on how these traits relate to the decision to use a SPAC (vs. a traditional IPO), rather than their relationship with performance (Kolb and Tykvová 2016).

Dimitrova was one of the few authors to revisit and explicitly consider SPAC performance determinants in an attempt to address the gap in the literature (Dimitrova 2017). Her exploration of cross-sectional variations in first-day and one-year returns led to the conclusion that the performance is worse when acquisitions occur near the end of the pre-determined two-year deadline and that sponsor involvement in the merged firm's governance improves the long-term outcomes. Further, she identified strong implicit incentives of SPAC sponsors as the key underlying determinant of performance. By this time, it had been demonstrated that sponsors acquire costly information on target firm characteristics at a significant effort; however, Dimitrova did not address the desirable or undesirable characteristics of target companies in her research (Chatterjee et al. 2014). To date, there still have not been any major efforts to investigate exactly what target firm features sponsors should evaluate.

In other portions of the literature, authors have focused on the role of SPAC sponsors, demonstrating that the usage of SPACs can be explained by sponsors acting as non-bank certification intermediaries that match investors' preferences (Bai et al. 2021). In addition to their core finding that SPAC volume is negatively related to market-wide uncertainty, Blomkvist and Vulanovic further expanded the role of the SPAC sponsor by identifying that sponsors can add investment credibility signals by purchasing additional warrants (Blomkvist and Vulanovic 2020). Gahng et al. (2021) suggests the purchase of additional warrants can be immensely profitable to sponsors, as their research found that SPAC warrant holders have profited the most in recent SPAC transactions, significantly more so than either IPO investors or common shareholders (Gahng et al. 2021).

This work addresses a major gap in the current literature by holistically considering which characteristics have a discernible impact on the market performance and should therefore be evaluated by SPAC sponsors. It builds upon prior SPAC performance determinant studies conducted by Murray and Dimitrova by considering market performance at the relevant sponsor lock-up timeframes, rather than only first-day returns. It additionally is one of the first studies to integrate a wide variety of features related to the target company, rather than just the SPAC or transaction into any type of SPAC performance determinant analysis.

### 2.2. Machine Learning Approaches for the Modeling Prediction of Financial Securities

While no published works to date have considered machine learning approaches explicitly for modeling SPAC performance, there is ample prior literature on the use of these techniques for predicting the performance of other market securities (Obthong et al. 2020). Early work in this area focused on using cluster algorithms to reduce dimensionality and noise in stock market data (Wang et al. 2006). This line of research continued for several years, eventually leading to the use of genetic algorithms to better capture non-spherical clusters in time-series data (Haviluddin and Alfred 2015). Genetic algorithms were a significant improvement on prior generations of clustering and were found to be particularly well-suited for demand forecasting applications (Wang et al. 2011).

Following the introduction of genetic algorithms, the literature saw a notable rise in publications using neural networks for predicting security performance, and the focus shifted away from clustering techniques to these new approaches (Göçken et al. 2016). Back-propagated and artificial neural networks were both used to model security predictions at similar accuracy levels; the former is noted in the literature as slower to converge, but also less prone to overfitting (Singh and Tripathi 2017; Yan et al. 2019). Generalized Regression

Neural Networks (GRNNs) were introduced in response to speed challenges associated with back-propagated and ANN algorithms, though they came at a cost of significant computational power.

On the opposite end of the computational spectrum, simple classification and regression tree (CART) models were demonstrated to capture non-linear security prediction problems well and provide easy-to-interpret results for understanding the importance of various features (Pradeepkumar and Ravi 2017). A key concern with CART models, however, is their instability when the training dataset changes (Razi and Athappilly 2005). Logistic regression models were demonstrated as a more robust approach for tackling non-linear problems without the same high computational demands associated with neural networks (Wu and Li 2018). Logistic regression, in particular, was shown by Mironiuc and Robu to be useful in identifying determinants of stock performance (Mironiuc and Robu 2013). Their analysis successfully demonstrated that by aggregating certain financial ratios for stocks on the Bucharest Stock Exchange, they could determine those metrics that were critically relevant to stock performance.

Random forest and long short-term memory (LTSM) models have gained popularity in recent years as some of the most common approaches for modeling the prediction of securities (Omar et al. 2022). Random forest offers the best advantages of decision tree and logistic regression by effectively combining the two techniques (Pradeepkumar and Ravi 2017). In contrast, LTSM takes its heritage from neural networks and is uniquely interesting in its ability to detect "hidden" patterns that are shared across securities (Selvin et al. 2017; Pradeepkumar and Ravi 2017).

With the digitalization of the accounting profession, businesses are changing how they approach financial data (Maričić et al. 2019). The increased use of intelligent methods in accounting is automating many routine tasks and enabling a wealth of data available to practitioners (Gulin et al. 2019). Despite this, the use of artificial intelligence models for visualization and predictive analysis is an under-researched area of the field (Cockcroft and Russell 2018). This work aims to demonstrate how a simple machine learning framework can be a powerful tool for identifying the relevant features related to market performance. The primary contribution in the machine learning setting is to add to the body of literature on intelligent methods in accounting with an additional application of how machine learning can inform financial management and decision-making.

## 3. Materials and Methods

To identify relevant SPACs for testing which SPAC/target company features are relevant to sponsors, five filtering criteria were applied to the CapitalIQ Transactions table:

- Industry Classification (Target/Issuer): Blank Checks (6770).
- Public Offerings Initial Filing Date: 1 January 2000 to 1 April 2021.
- All Transactions Closed Date: 1 January 2000 to 1 April 2021.
- Company Status (Target/Issuer): Acquired.
- IPO Exchange: Major US Exchanges (NYSE, NYSEAM, and NASDAQ).

These criteria were used to eliminate the prior generation of "Blank Check" companies (pre-2000), non-SPAC reverse mergers, and SPACs trading on off-market exchanges. In total, CapitalIQ returned 372 companies meeting the widely held definition of a SPAC outlined above. After data cleaning, the final sample included 310 SPACs trading on the NASDAQ, NYSE, or NYSEAM that IPOed and completed acquisitions between January 2000 and April 2021 with complete data. Of these, 130 had over one year of performance data and were included in the primary analysis. For each of these records, returns were compiled using the University of Chicago Center for Research in Securities Pricing (CRSP).

The independent variables for this study are a collection of SPAC, transaction, and target characteristics that can be influenced by SPAC sponsors. To ensure a comprehensive evaluation of potential features beyond what is currently seen in the literature, multiple data sources were aggregated for the source dataset. A combination of CapitalIQ and Compustat were used to isolate companies meeting the definition of a SPAC, and their

associated IPO data, which was then augmented with target company information from Pitchbook and data collected from SEC 425 Exhibit 99.x Investor Presentation filings.

For transaction characteristics, transaction ownership, and target venture involvement, a series of scale and binary indicator variables were collected for each SPAC, which then underwent dimensionality reduction via the popular k-means clustering technique. Clustering techniques have a long history of use in the literature and it was applied here to reduce potential of multi-collinearity between the extremely interrelated decision variables, such as those seen in transaction sources and uses (Sarlin 2015; Yuan and Yang 2019). Clusters were developed based on the entire dataset (n = 310)—their exact groupings and centers can be found in Appendix A.

As seen in Table 1, a total of 12 individual decision variables and 4 clustered decision variables related to SPAC, target, and transaction characteristics were included as independent variables (i.e., potential SPAC performance determinants) in this study.

**Table 1.** Independent variables included in the analysis.

| Independent Variable | Type | Description |
|---|---|---|
| Nasdaq | Binary | Traded on NASDAQ |
| Serial Issuer [1] | Binary | Repeat SPAC sponsor |
| Early Announcer | Binary | Announced < = 180 days after IPO |
| Late Announcer | Binary | Announced > 640 days after IPO |
| Majority Top-tier Underwriters | Binary | Majority of underwriters from Tier 1/2 banks [2] |
| Target Domestic [3] | Binary | US-based corporate headquarters |
| Target Corporate/Private Equity-backed [3] | Binary | Prior corporate or private equity ownership |
| Year Fixed Effect | Binary | Indicates if SPAC closed acquisition post-2016 |
| Demonstrated Profitability [4,5] | Binary | Disclosed positive profitability metric |
| SPAC Size | Scale (Log) | Intended SPAC size at IPO |
| Target Age [3] | Scale (Log) | Company age at the time of announcement |
| Announced Enterprise Value [5] | Scale (Log) | Implied enterprise value in the investor presentation |
| Venture Involvement Cluster [3,5,6] | Categorical | Aggregation of venture funding; valuation step-up |
| Revenue Projections Cluster [5,6] | Categorical | Aggregation of projection length, CAGR, and year 0 revenue |
| Sources and Uses Cluster [5,6,7] | Categorical | Aggregation of cash sources and uses |
| Ownership Cluster [5,6] | Categorical | Aggregation of public, PIPE, sponsor, and existing ownership |

[1] Classified from Compustat; repeated, sequential names classified as serial issuers, [2] includes Credit Suisse, Citi, Goldman Sachs, Deutsche Bank, Morgan Stanley, UBS, BofA, J.P. Morgan, and Barclays, [3] as identified by Pitchbook; [4] includes gross profit, net profit, EBITDA, or adjusted EBITDA; [5] at time of announcement/displayed in investor presentation; [6] cluster variables developed using k-means clustering, [7] includes: (uses) cash in trust, cash on balance sheet, PIPE equity, roll-over equity/issuance of new shares, new/existing debt, sponsor shares, and other; (sources) roll-over equity/stock consideration, cash to balance sheet, cash consideration, debt repayment, fees and expenses, sponsor shares, and other.

As there is limited prior research on the potential determinants of market performance for SPACs, this study seeks a broad screen of potentially influential variables. Machine learning models are a useful tool for such a problem, as they are well-suited for modeling non-linear trends, especially when there is little understanding of the true relationship between independent and dependent variables (Wang et al. 2011).

A variety of approaches for feature selection exist across machine learning models in the literature; however, the biomedical field takes the most stringent stance on feature inclusion/omission due to their unique clinical applications (Tolles and Meurer 2016; Miao and Niu 2016). For biomedical purposes, only variables that are clinically important (i.e., robust functional justification can be made for their inclusion) and statistically significant should be included in the final model (Zhang et al. 2018). As the purpose of this study is to identify true performance determinants of SPACs, not just good model predictors, it follows a similarly strict methodology to identify only those features that possess the most prominent relationship with performance.

To achieve this goal, this study analyzes which predictor variables are used to classify SPAC market performance across several popular machine learning models. The dependent variable across all models is a binary classifier indicating whether the SPAC outperformed their exchange during the first 12 months since the transaction close. The classifier approach was chosen instead of using the actual stock price to capture those SPACs that may have generated "poor" returns but beat the performance of the overall market during their first year of trading.

The methodology assumes a "buy-and-hold" strategy for both the SPAC and exchange at the time of the transaction close and evaluates whether the return from the SPAC was more than that of the exchange over the same period. The simplest interpretation of this variable is whether the SPAC is an exchange under- or overperformer at 12 months post-transaction close. The exact calculation for this classifier is seen below:

$$Market\ Adjusted\ Return = \frac{Log(Price_{SPAC,t}) - Log(Price_{SPAC,C})}{Log(Price_{SPAC,c})} - \frac{Log(Price_{Exchange,t}) - Log(Price_{Exchange,C})}{Log(Price_{Exchange,c})}.$$

where $t$ = transaction close plus 12 months and $c$ = transaction close. In the case of the SPAC, an assumed share price of USD 10 (nearly all SPACs in the last decade list at a stated offer price of USD 10) is used instead of the actual security price at transaction close (Gahng et al. 2021). Log returns were used due to the stochastic behavior of all log-return time-series data—the advantage being that relative changes are easier to spot and it is possible to compare across variables of a different scale (Dhifaoui 2022).

Each model is trained on a subset of the sample—"Training" data—to identify general predictors, and is then further validated on a hold-out sample—"Testing" data. For SPACs with 12 months of data, a total of 130 observations are included in the analysis; n = 87 in the training set and n = 43 in the testing set. Potential performance determinants are identified by comparing variable significance, frequency, and impact across machine learning models. For this analysis, three common machine learning models were chosen: (1) classification and regression tree, (2) logistic regression, and (3) LASSO regression. Each of these algorithms has demonstrated a history of use in the literature for predicting market performance and is notably less prone to overfitting than other techniques (Obthong et al. 2020). As compared to neural network models, these algorithms can be used to effectively make predictions with extremely small datasets and provide clear visibility into their feature set when interpreting results (Pradeepkumar and Ravi 2017).

Gradient boosting and random forest algorithms are also a natural extension of CART and logistic regression techniques that have been shown to increase classification accuracy in market performance predictions (Omar et al. 2022). In the context of this research, however, the effort is distinctly different from much of the prior machine learning literature on securities predictions, which most often seek to optimize model accuracy. Rather, the primary concern here is to identify those predictors that can be consistently relied upon by SPAC sponsors when forming investment theses. The chosen methodology was selected in an attempt to isolate only the most impactful of traits, as is often seen in clinical research (Shipe et al. 2019). As such, the approach avoids these more complex gradient boosting and random forest models in favor of more easily interpretable, foundational algorithms.

## 4. Sample Trends and Variable Characterization

Within this sample, the post-merger SPAC performance is highly divergent. Across 130 observations, the 12-month buy-and-hold returns range from −96% to +398%, as seen in Figure 1. Extreme outliers in the form of overperformers generating 300%+ during their first year suggest why risk-loving investors are drawn to this asset class despite the overall performance.

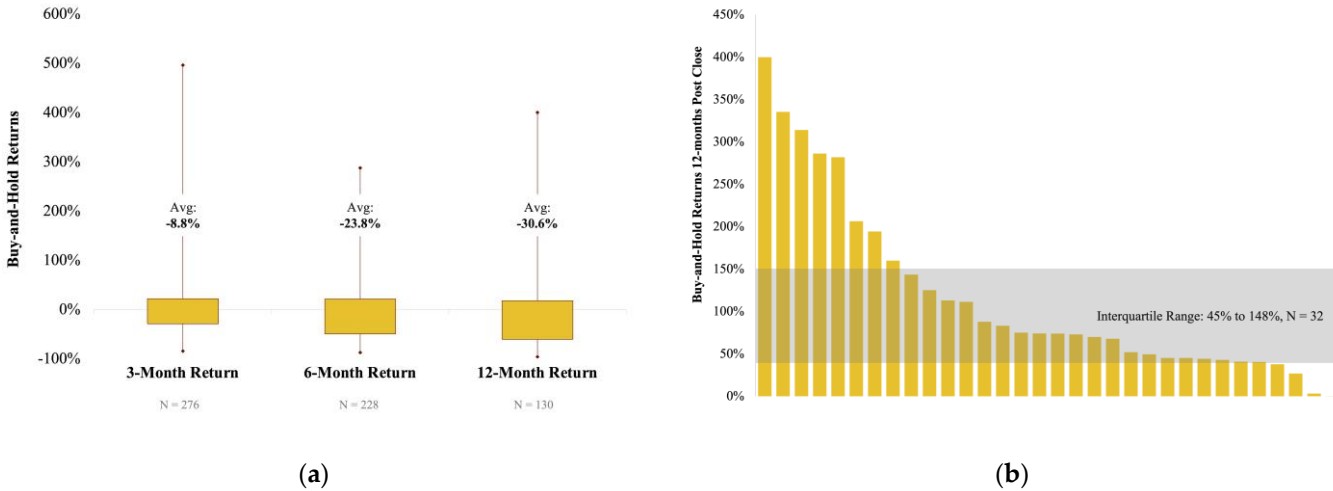

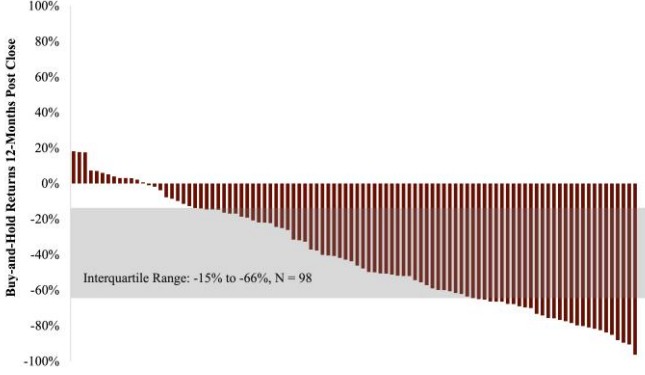

(c)

**Figure 1.** Sample trends—SPAC post-merger performance summary. (**a**) Buy-and-hold returns of SPACs at 3, 6, and 12 months post-transaction close; (**b**) SPAC market overperformer returns at 12 months post-transaction close; and (**c**) SPAC market underperformer returns at 12 months post-transaction close.

Segmenting by market under- and overperformers shows an even more stark delineation in returns, with overperformers delivering a median return of +74% at 12 months post-close, while underperformers delivered a median return of –45% during the same time. The interquartile range of returns was –15% to –66% for underperformers, and +44% to +148% for overperformers. Within this sample, only ~25% of SPACs outperformed the market at 12 months post-close (n = 32), while the remaining 75% underperformed (n = 98). Note that the underperformers did not necessarily generate a negative return and the overperformers did not necessarily generate a positive return, their return was simply greater than that of the SPAC's exchange during the same period.

Given this large class imbalance between market under- and overperforming SPACs, it should be noted that this study does not seek to address this imbalance using either the Support Vector Machine (SVM) or Synthetic Minority Oversampling (SMOTE), as is sometimes seen in the literature (Ramyachitra and Manikandan 2014). These types of pre-processing approaches are used as a means of artificially expanding the dataset so that the predictors are more pronounced and are most commonly used in direct-response modeling, where the primary class often contains <2% of observations (Rogić et al. 2022) As compared to digital marketing, this effort has an order-of-magnitude higher level of class balance, and the need for such magnification of trends is far smaller. Further, the small

sample size presents additional problems for resampling, cross-validation, and interference techniques, which have been shown to bias the algorithm classification accuracy in datasets up to n = 1000 (Vabalas et al. 2019).

Figures 2–4 detail independent variable trends related to the SPAC, target, and transaction characteristics. As seen here, there is no "one-size-fits-all" SPAC—they come in many sizes, from many industries, and with vastly different prior capitalizations. Exponential growth in SPACs closing acquisition began in ~2015, surging as sponsors sought to bring new venture-backed firms to the market. As seen in Figure 2, growing popularity in the USD 100–USD 300 M and larger SPAC sizes has attracted high-caliber underwriters, and a greater portion of SPACs are debuting on the NYSE, rather than the Nasdaq.

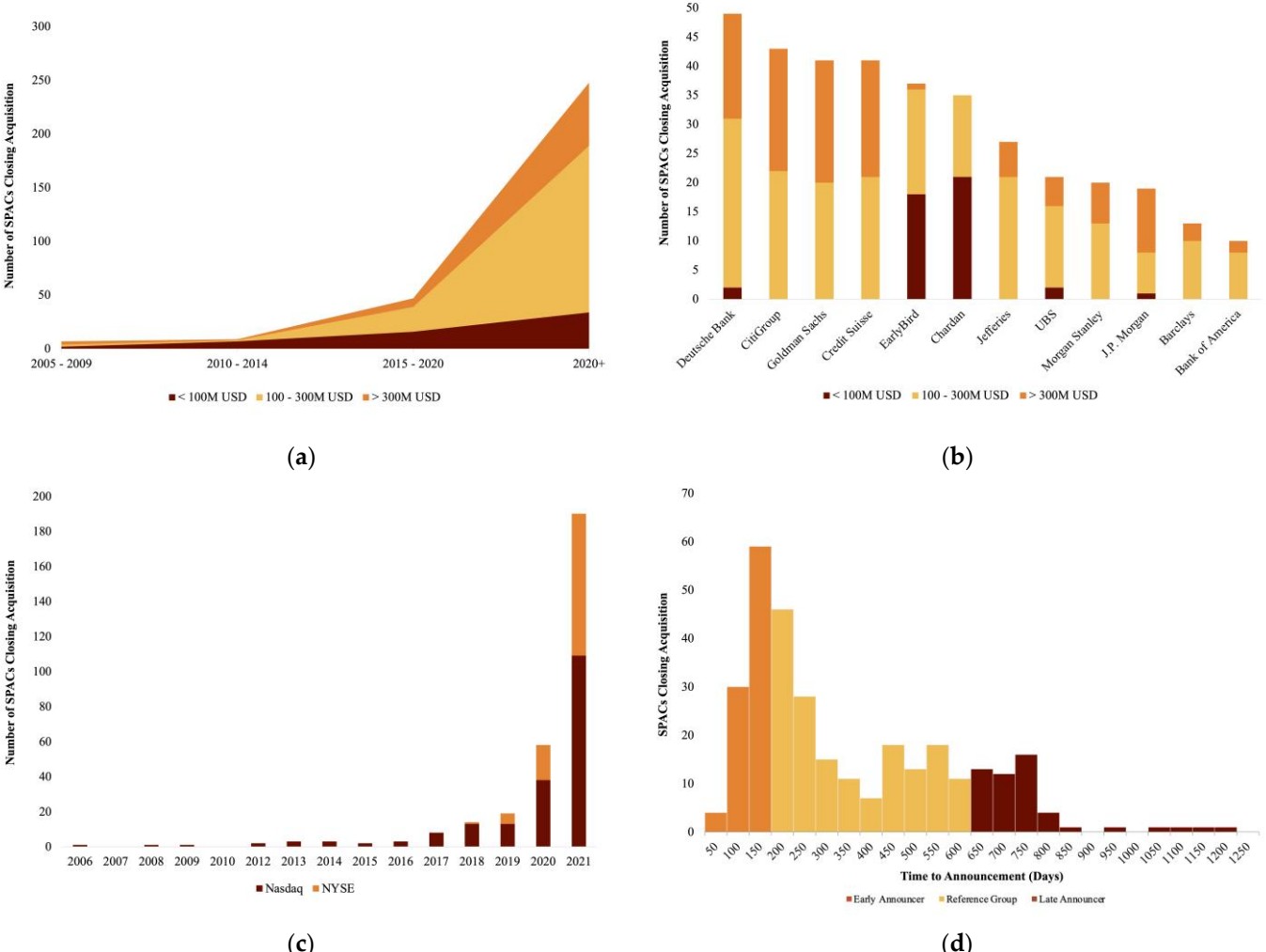

**Figure 2.** Sample trends—SPAC characteristics. (**a**) Evolution of SPAC sizes at the time of IPO; (**b**) SPAC underwriters and associated SPAC sizes; (**c**) SPAC exchange popularity over time; and (**d**) distribution of time-to-announcement for SPACs closing acquisition.

In terms of the target companies that SPACs eventually take public, SPACs are most commonly used for companies in the services and manufacturing sectors—specifically, pharmaceuticals, software, computer programming, and data processing encompass the plurality of SPACs in the sample. As seen in Figure 3, the firms have historically been domestic and are often younger (<20 years old), corroborating the findings from Ma, Bai, and Zheng.

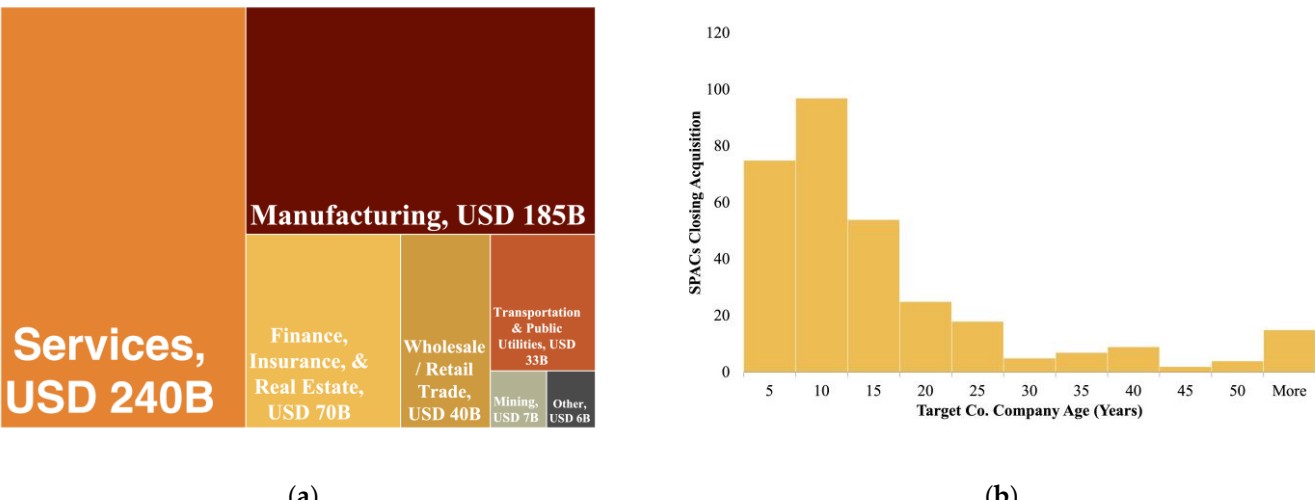

**(a)** **(b)**

**Figure 3.** Sample trends—target company characteristics. (**a**) Target company announced enterprise value by industry. (**b**) Distribution of target company age at the time of acquisition.

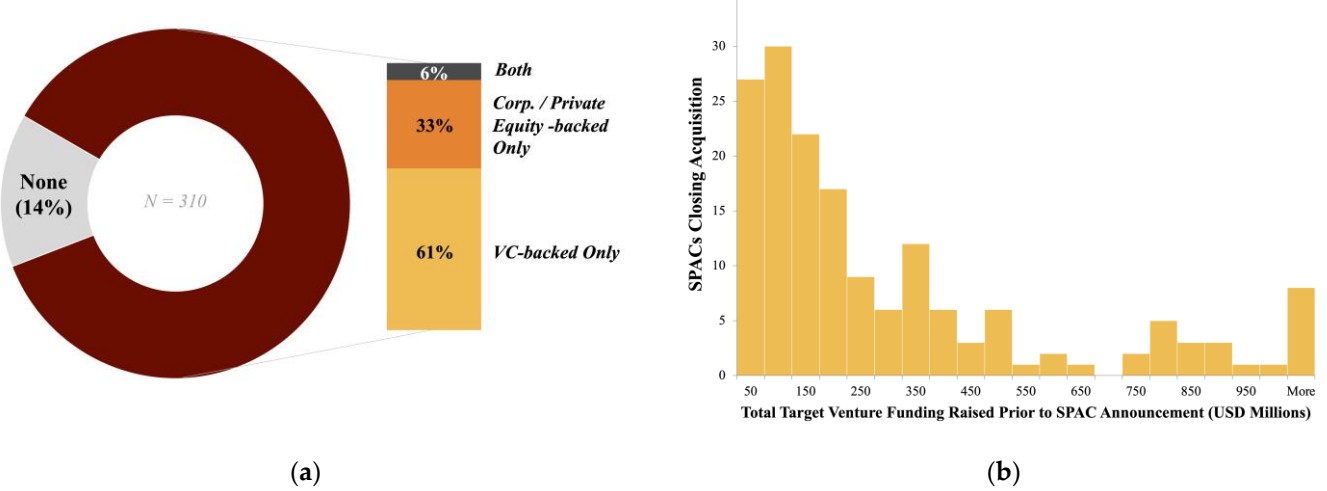

**(a)** **(b)**

**Figure 4.** Sample trends—target company venture capital involvement. (**a**) Target company venture capital/private equity involvement as a percentage of the entire sample. (**b**) Distribution of target company venture capital raised before announcement (VC-backed target companies only).

The majority of these firms have some type of prior investor involvement (corporate, private equity, or venture capital)—around 86% of the sample. Before 2016, far fewer SPAC targets had venture, corporate, or private equity backing, but the dramatic increase in SPACs (particularly SPACs with venture-backed targets) over the last 5 years skews the entire sample toward heavy prior investor involvement. For those firms that do have prior investor involvement, few have raised more than USD 300 M before the announcement.

Figure 5 highlights the growing use of roll-over equity as a transaction source of cash, as well as the increasing popularity of PIPE investors in SPAC transactions. In this sample, the majority of SPAC transaction proceeds usage over the past six years have been roll-over equity compensation to new or existing owners (>40% of cash sources), followed by cash considerations and debt repayment (~30%). Since 2016, only around 30% of transaction proceeds have gone to the merged entity balance sheet within this sample. Finally, panel (d), profiles recent trends suggesting declining levels of public (SPAC) ownership in the merged entity in favor of PIPE and existing shareholders.

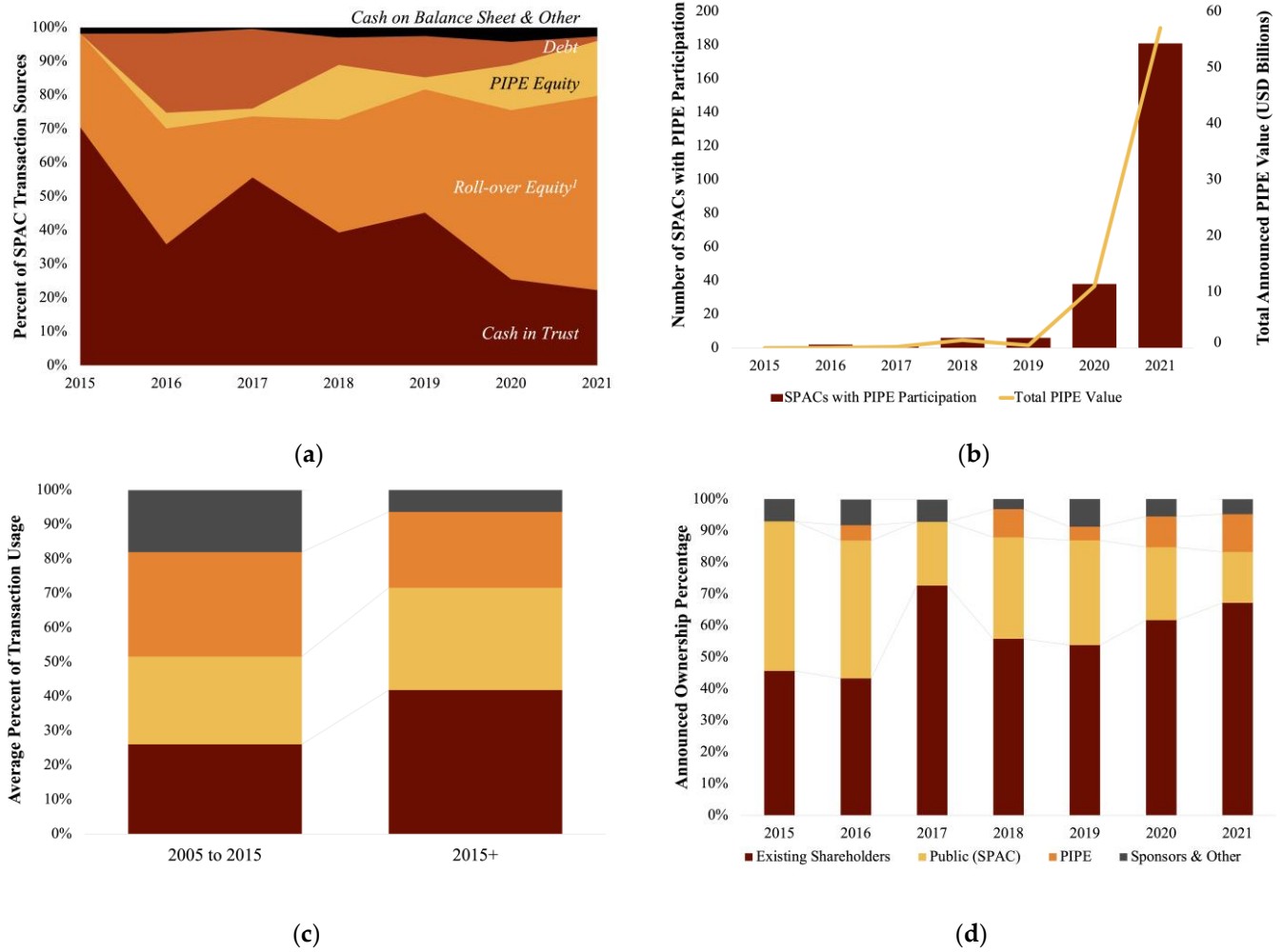

**Figure 5.** Sample trends—SPAC transaction characteristics. (**a**) Announced average transaction sources by close year. (**b**) Announced PIPE participation by close year. (**c**) Announced average transaction uses by close year. (**d**) Announced average transaction ownership composition by close year.

## 5. Results

### 5.1. Classification and Regression Tree Results

The first machine learning model used to assess the potentially relevant features for SPAC sponsors to evaluate was a classification and regression tree (CART). The results of this model indicate that corporate/private equity backing, enterprise value, and ownership structure are most predictive of market performance 12 months post-transaction close, as seen in Table 2 and Figure 6.

**Table 2.** Classification and regression tree model parameters and results.

| Model | (1) | (2) |
|---|---|---|
| Model Type | CRT Unpruned | CRT Pruned |
| Estimate | 0.172 | 0.172 |
| Standard Error [1] | 0.040 | 0.040 |
| Out-of-sample f-Score | 0.32 | 0.32 |
| Number of Terminal Nodes | 4 | 3 |
| Average Observations per Terminal Node | 22 | 29 |

[1] No minimum distance in error specified; tree with the smallest risk selected.

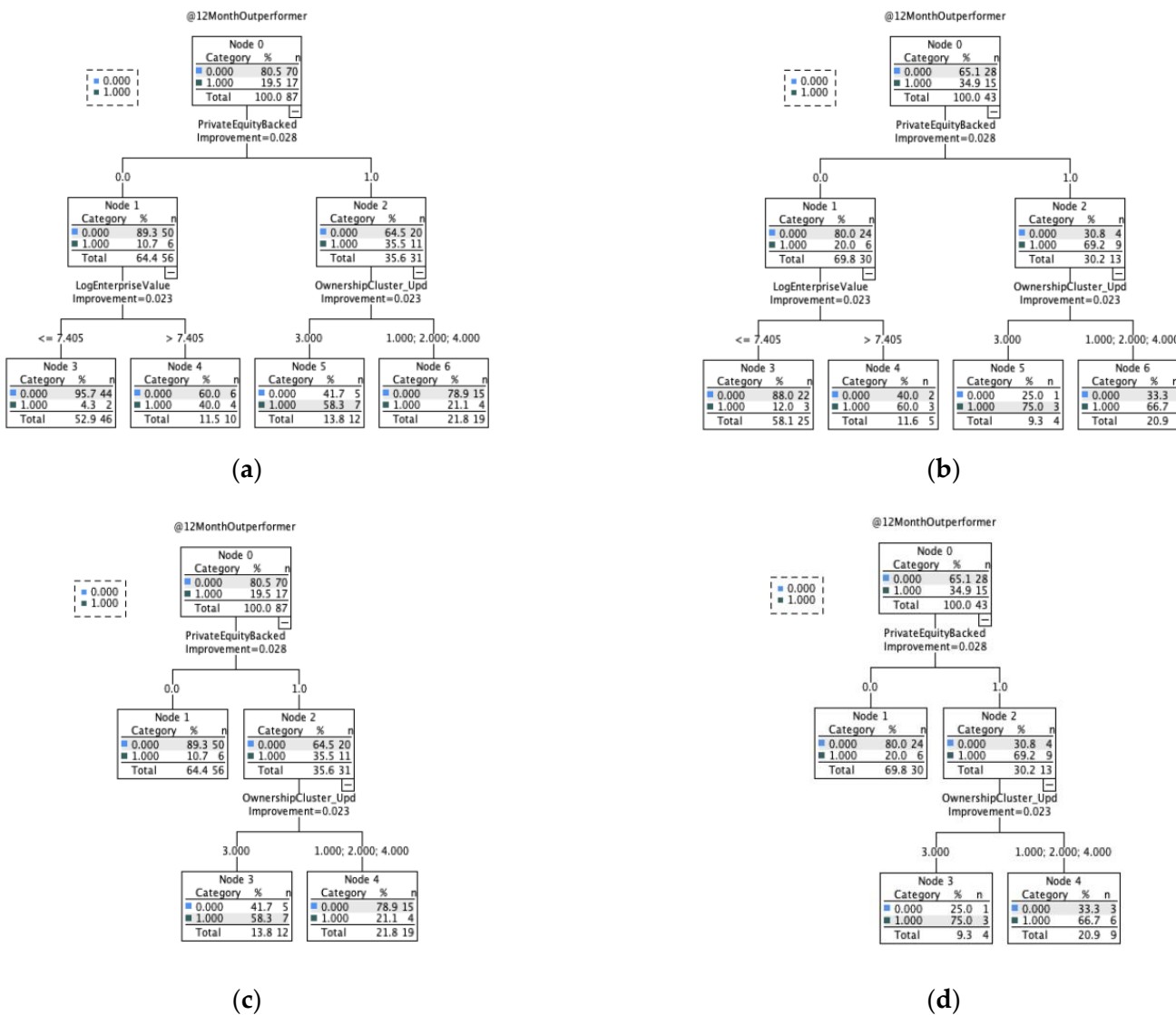

**Figure 6.** CART decision tree results. (**a**) Unpruned CART model decision tree results for training sample. (**b**) Unpruned CART model decision tree results for testing sample. (**c**) Pruned CART model decision tree results for training sample. (**d**) Pruned CART model decision tree results for testing sample.

The initial data splits well on corporate/private equity backing, with non-backed firms generally resulting in underperformance (50 out of 56 in the training sample). Further splits are made on enterprise value and ownership structure, both of which improve the classification of underperformers, but not overperformers. The pruned tree retains the ownership structure split but drops the enterprise value consideration while maintaining an identical out-of-sample standard error and predictive performance.

As seen by the out-of-sample f-score in Table 2, and the expanded confusion matrix in Figure 7, the overall classification accuracy of the CART models is quite poor. The classification of market underperformers was completed with a 92.9% accuracy; however, the 58.8% false-negative rate suggests that this was due to the model nearly always predicting an underperformance. The model identified overperformers with a 41.2% true-positive rate (correct overperformer predictions) and only 7.1% false positives (incorrect overperformer predictions), further indicating that few observations (correctly or incorrectly) were classified as overperformers.

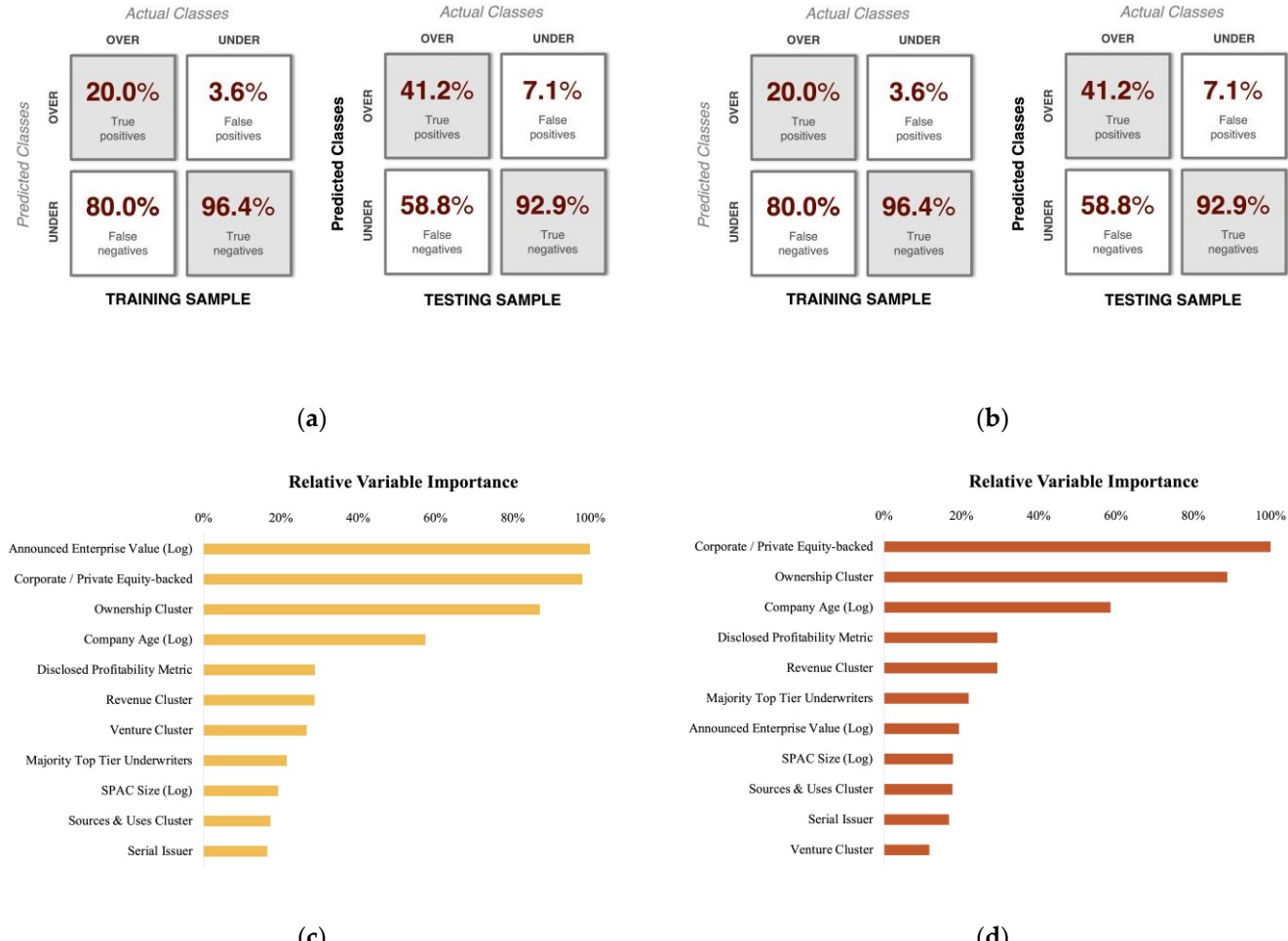

**Figure 7.** Classification and regression tree model performance. (**a**) Unpruned CART model performance results—confusion matrix (training and testing samples). (**b**) Unpruned CART model normalized variable importance results. (**c**) Pruned CART model performance results—confusion matrix (training and testing samples). (**d**) Pruned CART model normalized variable importance results.

As noted previously, the pruned tree had an identical out-of-sample predictive performance (Figure 7); however, it did result in differing levels of variable importance. In the final pruned tree, corporate/private equity backing, ownership structure, company age, disclosed profitability metrics, and revenue projections all had a variable importance > 20% and should be considered as potentially influential features for SPAC sponsors to investigate. The low accuracy of this algorithm in identifying and distinguishing classes, however, suggests that these feature selection results should be interpreted cautiously.

*5.2. Logistic Regression Results*

The initial logistic regression model developed for this effort suffered from multicollinearity caused by the overlapping of independent variables. This is to be expected given the small sample size, and it was addressed by sequentially omitting variables with the highest standard error, as suggested in the literature (Senaviratna and Cooray 2019). This resulted in the eventual omission of nine independent variables from the logistic regression model that were captured in the CART analysis: domestic, post-2016, revenue cluster 2, sources and uses clusters 1 and 4, ownership clusters 1, and venture clusters 3 and 4. The primary impact of these omissions is a widening of the reference (null) group.

Despite the omissions, the forward stepwise logistic regression greatly improved the model accuracy, with an f-score of 0.64 in the testing dataset. Similar to the CART models,

the forward stepwise model isolated corporate/private equity backing and SPAC size as the primary predictors of market performance (Table 3). Based on this model, the presence of corporate/private equity backing in the target company increases the odds of market outperformance by 5× The SPAC size, in contrast, appears to have an inverse relationship with the market outperformance (i.e., larger SPACs are less likely to outperform). Drawing on the results of the CART diagrams in Figure 6, however, it can be inferred that this predictor is primarily used to help isolate the few non-corporate/private equity-backed overperformers and may not be reliable as a standalone predictor.

**Table 3.** Forward stepwise logistic regression results.

| Variables in Equation | B | S.E. | Sig. | Exp(B) |
|---|---|---|---|---|
| Log SPAC Size | −0.470 | 0.085 | <0.001 *** | 0.666 |
| Corporate/Private Equity-backed | 1.613 | 0.598 | <0.001 *** | 5.017 |

*** $p < 0.01$; model overall out-of-sample f-score: 0.64.

An additional logistic regression model without stepwise selection was also developed, though it only marginally improved the classification accuracy to an f-score of 0.67. The classification improvements, seen in Figure 8, primarily stemmed from the higher number of true negatives and lower number of false positives, suggesting that this model was better able to distinguish edge cases. More interestingly, this model leveraged several additional predictor variables to classify the SPAC performance, as seen in Table 4.

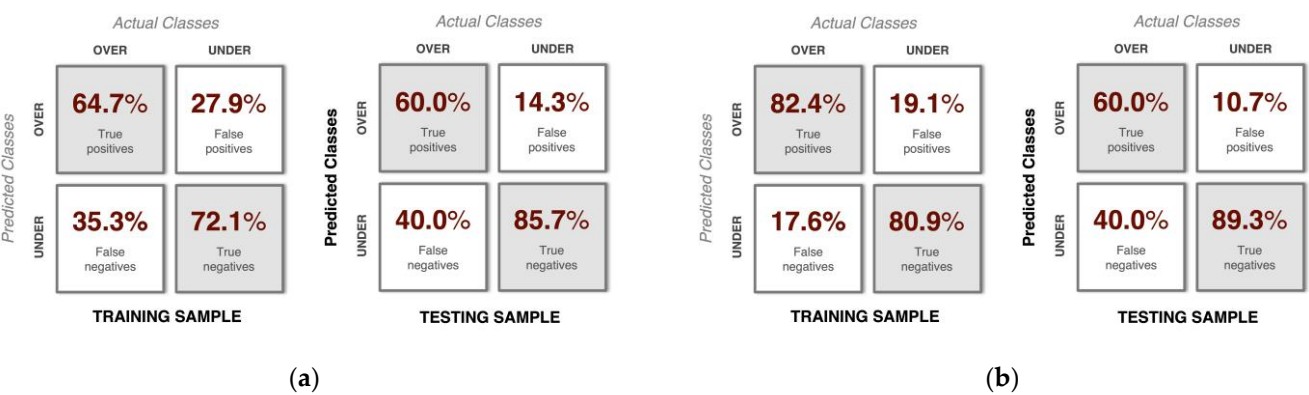

**Figure 8.** (**a**) Forward stepwise logistic regression model performance results—confusion matrix (training and testing samples). (**b**) All variable logistic regression model performance results—confusion matrix (training and testing samples).

In addition to corporate/private equity backing, this model identified early/late announcer, disclosed profitability, ownership clusters 2 and 4 (concentrated public and existing shareholder ownership), and venture cluster 2 (venture-backed companies with >USD 300 M completing a fundraising round within the last 6 months) as potential performance determinants. Of these predictors, the venture cluster 2 had the largest impact on the odds of market outperformance—an increase of 58×—which was even greater than that of corporate/private equity backing. Note that due to the inclusion of so many independent variables in this model, the reference group is quite narrow and the coefficient values should be interpreted cautiously.

**Table 4.** All variables logistic regression results.

| Variables in Equation | B | S.E. | Sig. | Exp(B) |
|---|---|---|---|---|
| Nasdaq | −0.717 | 0.884 | 0.417 | 0.488 |
| Log SPAC Size | −0.828 | 0.856 | 0.333 | 0.437 |
| Serial Issuer | −0.411 | 1.026 | 0.689 | 0.663 |
| Early Announcer | 2.289 | 1.402 | 0.102 | 9.868 |
| Late Announcer | 1.910 | 1.030 | 0.064 * | 6.754 |
| Majority Top-tier Underwriters | 0.267 | 1.005 | 0.791 | 1.306 |
| Log Age | 0.656 | 0.514 | 0.202 | 1.926 |
| Corporate/Private Equity-backed | 2.831 | 1.281 | 0.027 ** | 16.956 |
| Disclosed Profitability Metric | 2.799 | 1.631 | 0.086 * | 16.429 |
| Log Announced Enterprise Value | 0.103 | 0.633 | 0.871 | 1.109 |
| Revenue Cluster 1 | −3.112 | 1.672 | 0.063 * | 0.045 |
| Revenue Cluster 3 | −2.585 | 1.926 | 0.180 | 0.075 |
| Revenue Cluster 4 | −2.904 | 1.834 | 0.113 | 0.055 |
| Sources and Uses Cluster 2 | 0.383 | 1.442 | 0.791 | 1.466 |
| Sources and Uses Cluster 3 | −0.326 | 1.108 | 0.768 | 0.722 |
| Ownership Cluster 2 | −2.594 | 1.556 | 0.096 * | 0.075 |
| Ownership Cluster 3 | −0.347 | 1.138 | 0.760 | 0.706 |
| Ownership Cluster 4 | −3.060 | 1.578 | 0.052 ** | 0.047 |
| Venture Cluster 1 | 1.325 | 1.275 | 0.299 | 3.762 |
| Venture Cluster 2 | 4.069 | 1.997 | 0.042 ** | 58.491 |

\* $p < 0.10$, \*\* $p < 0.05$; model overall out-of-sample f-score 0.67.

### 5.3. LASSO Regression Results

An alternative approach to feature selection in machine learning is to use a ridge or LASSO regression as a means of narrowing down the relevant predictors (Bhadra et al. 2019). The LASSO technique is particularly useful in this case, as it is designed for selection rather than shrinkage—i.e., it can set coefficients to zero, effectively removing "unimportant" variables from the model. For this research, it is applied to isolate only those predictors that are most critical to the SPAC performance. No training/testing partitions were implemented for this model, and the lambda parameter was set to a maximum of 0.5 (increment of 0.01). Validation was performed using a 0.632 bootstrap with a minimum amount of resampling (100 samples), and the resulting model demonstrated an adjusted r-square of 0.149.

As seen in Table 5, the LASSO omitted all but eight variables from the final model. Of these, only corporate/private equity backing was a consistently reliable predictor across the entire bootstrap. The odds impact of corporate/private equity backing can be interpreted similarly to the logistic regression—prior involvement increases the odds of market outperformance by 1.27×. Note that this coefficient is substantially smaller than that seen previously (5× and 17× in the prior logistic models). The smaller coefficient magnitude is representative of a wider reference group due to fewer independent variables and a more refined estimate of the impact on market performance. The additional variables included in the model were not omitted, but they were also not significant, suggesting they have at least some influence on market performance in combination with corporate/private equity backing, but they may not be reliable by themselves.

**Table 5.** LASSO regression results.

| Variables in Equation | B | S.E. | Sig. | Exp(B) |
|---|---|---|---|---|
| Log SPAC Size | −0.003 | 0.055 | 0.955 | 0.997 |
| Early Announcer | 0.098 | 0.081 | 0.237 | 1.103 |
| Late Announcer | 0.016 | 0.048 | 0.903 | 1.016 |
| Majority Top-tier Underwriters | 0.047 | 0.064 | 0.590 | 1.048 |
| Corporate/Private Equity-backed | 0.240 | 0.099 | 0.004 *** | 1.271 |
| Disclosed Profitability | 0.028 | 0.052 | 0.742 | 1.028 |
| Log Announced Enterprise Value | 0.034 | 0.074 | 0.812 | 1.035 |
| Venture Cluster 2 | 0.128 | 0.093 | 0.159 | 1.137 |

*** $p < 0.01$.

### 5.4. Supplemental Linear Regression Results

Across the CART, logistics, and LASSO regression, corporate/private equity backing in the target firm was the only consistent, statistically significant predictor for classifying market performance, suggesting that beyond being a good predictor, it may be a true determinant of SPAC performance. Prior literature in the financial context has traditionally relied upon linear regression modeling with fixed effects for demonstrating performance-determinant relationships (Wang et al. 2021). To further validate the results and conclusions being drawn from the above machine learning models, an additional, traditional model was developed to test the importance of this variable after the study.

For this experiment, the dependent variable is log returns at 12 months (not the binary classifier), and the independent variable is corporate/private equity backing, as suggested by the machine learning models. The results of the linear model, as seen in Table 6, demonstrates that corporate/private equity backing has a statistically significant relationship with 12-month buy-and-hold returns for SPACs, even when controlling for year and industry fixed effects. The adjusted r-square value of 0.099 indicates that approximately 10% of the variation in 12-month returns for SPACs can be explained by corporate/private equity backing and fixed effects. While this simplified model leaves a great deal of variance unexplained, it corroborates the results of the machine learning models and further suggests corporate/private equity backing as a performance determinant for special purpose acquisition companies.

**Table 6.** Linear regression results.

| Model | (1) | (2) | (3) |
|---|---|---|---|
| N | 130 | 130 | 130 |
| Adjusted R-Square | 0.088 | 0.081 | 0.099 |
| Constant | −0.554 ** | −0.540 ** | −0.532 ** |
| Corporate/Private Equity-backed | 0.557 *** | 0.558 *** | 0.630 *** |
| Year Fixed Effects | - | Yes | Yes |
| Industry Fixed Effects (1-digit SIC) | - | - | Yes ** |

** $p < 0.05$, *** $p < 0.01$.

## 6. Discussion

### 6.1. Identifying Features of High-Performing SPACs Using Machine Learning

The aim of this effort was to identify the features of SPACs and their target companies that are consistently related to market performance at sponsor lock-up windows. By comparing the predictor variables used by the machine learning models in this study, sponsors can see which features are relevant to future market performance and should be evaluated during their search process. In this case, variables that appeared across the models indicate that they can be reliably used, in conjunction with other variables, to classify SPAC performance, while variables that appeared as statistically significant across the models indicate that they can be reliably used as a standalone predictor to classify SPAC performance. In the context of SPAC sponsors, the implication is that standalone predictors,

by themselves, may be enough to ensure market performance, while other predictors may be only one component of a successful transaction. Table 7 details the full frequency of occurrence for different independent variables in the machine learning models used for this analysis. For the all-variables logistic, only variables with *p*-values < 0.10 are included.

**Table 7.** Most common machine learning features used for predicting SPAC market performance.

| Model | (1) Unpruned CRT | (2) Pruned CRT | (3) Fwd. Stepwise | (4) All Variables Logit | (5) LASSO |
|---|---|---|---|---|---|
| Announced EV | X | - | - | - | X |
| Corporate/Private Equity-backed | X | X | X *** | X ** | X *** |
| Ownership Cluster (Any) | X | X | - | X ** | X |
| Company Age | - | - | - | - | - |
| Disclosed Profitability | - | - | - | X * | X |
| Revenue Cluster | - | - | - | X * | - |
| Venture Cluster (Any) | - | - | - | X ** | X |
| Major Top-tier Underwriters | - | - | - | - | X |
| SPAC Size | - | - | X *** | - | X |
| Sources/Uses Cluster (Any) | - | - | - | - | - |
| Serial Issuer | - | - | - | - | - |
| Early Announcer | - | - | - | X | X |
| Late Announcer | - | - | - | X* | X |
| Out-of-sample f-score | 0.32 | 0.32 | 0.64 | 0.67 | - |

* $p < 0.10$, ** $p < 0.05$, *** and $p < 0.01$.

In this study, the only reliable standalone predictor across all five models was the target company's corporate/private equity involvement. The frequency of appearance and consistent statistical significance across models for corporate/private equity backing suggest that it is a critically important feature of SPACs that outperform their exchange during the first 12 months of trading. Regression coefficients from the various machine learning models indicate the impact of corporate/private equity backing on the odds of market outperformance ranging from $+1.3\times$ to $+17\times$, depending on control factors and the reference group. The results from the traditional linear regression experiment further corroborate these results, suggesting that even when controlling for the year and industry fixed effects, corporate/private equity-backed SPACs generate returns of approximately +10% 12 months post-close.

To be precise about the exact definition of this variable, corporate/private equity-backed firms were classified based on Pitchbook's deal history tables. Firms that had a prior deal type of (1) buyout/LBO, (2) PE growth/expansion, or (3) a secondary private transaction involving a private equity firm were counted as having prior corporate/private equity backing. Additionally, firms that listed private equity firms as the primary owner of a "roll-up" company in their SPAC investor presentations from SEC Exhibit 99.x were also categorized as corporate/private equity-backed. From this definition, the variable should be broadly interpreted as firms that previously had private equity or corporate ownership, conducted a private equity-led growth or expansion fundraising round, or experienced a secondary market transaction where a private equity investor purchased a portion of their company.

This finding is particularly novel, as it suggests that features of the target firm can be determinants of market performance, in addition to characteristics of the SPACs, transactions, and market identified in prior research. As for why these firms outperform the market, it is unclear whether it is due to "Selection" or "Treatment" by the corporate/private equity investors (Bengtsson and Sensoy 2011). Given the above definition, it is clear that not all of these firms were fully owned by corporate/private equity investors, but rather, they had

prior involvement. In this context, "Treatment" explanations seem less likely—without full control, these professional investors would not have the high degree of influence required to make governance changes or shape strategic direction. With such a high proportion of target firms that are younger and venture-backed, a more logical explanation may be that corporate and private equity firms serve as credibility signals for sponsors that the targets are less risky and more capable of performing on public markets.

Beyond corporate/private equity backing, it can also be concluded that the announced enterprise value, early/late announcement, venture backing (particularly significant prior VC involvement > USD 300 M), and ownership structure may also be shared traits of SPACs that outperform their exchange. As these variables frequently appeared across models but did not show consistent statistical significance, it is unclear whether they simply help dial in the impact of corporate/private equity backing, or whether they are useful as standalone predictors. Additional testing is necessary to better understand the relationship between these variables and SPAC performance.

### 6.2. Discussion of Possible Study Biases and Their Impact

Sampling Bias—Machine learning models are only as good as their inputs, and sampling bias is a common concern for these types of studies. To mitigate sampling bias and model overfitting, training and testing data partitions were used throughout the study, and conclusions were only drawn from out-of-sample prediction results. To further demonstrate the applicability of these findings beyond the current data sample, an additional test was performed on a purist out-of-sample data tranche of SPACs reaching 12 months of performance data between 1 March 2022 and 23 May 2022. In total, this new data tranche contained 15 data points, and the results of the out-of-sample test are displayed in Table 8. As seen here, the model correctly identified two out of three under and overperformers out-of-sample, improving the overperformer classification accuracy and seeing only a marginal decrease in the underperformer classification accuracy. The results of this test suggest that, if there was sampling bias, it did not result in an overfit model.

**Table 8.** Purist out-of-sample model accuracy results.

| Model Performance | Sample Size | True Positives | False Positives | True Negatives |
|---|---|---|---|---|
| In-sample | 130 | 81.3% | 22.7% | 77.3% |
| Out-of-sample | 15 | 66.7% | 33.3% | 66.7% |

Selection Bias—Of the 372 SPACs identified from CapitalIQ, 62 were omitted from the sample. Of these, 45 records were found to have either not completed a merger or had otherwise incomplete identifying information. An additional 17 were found to not actively trade on major US exchanges (instead trading on TSX or OTC) and were also omitted from the sample. Alternative paid data sources, such as the SPAC Insider, suggest that as many as 400+ SPACs may have closed acquisitions between 2000 and 2021. These discrepancies and omissions underscore that the results shown here are only a sample of SPAC performance and should not be interpreted as population statistics. Based on the analysis of the omitted data points, as well as cross-comparisons with alternative data sources, there does not appear to have been any systematic omission of SPACs based on any of the dependent or independent variables in the study; given the small number of impacted datapoints, these omissions are unlikely to have an extreme impact on the study's findings.

Interpretation Bias—An assumed share price of USD 10 was used for all SPACs at the time of transaction close, reflecting the value to the SPAC sponsor rather than the value to a public investor. Because of this, readers may be biased to misinterpret the dependent variable as being reflective of whether the SPAC generated a greater return than the exchange for *investors* between transaction close and 12 months post-close, when in reality, these should be interpreted as the SPAC generating a greater return than the exchange for *sponsors*. Additionally, the collection of many data points from the initial SEC Exhibit

99.x investor presentations at the time of announcement creates a unique interpretation for the results of this study: all independent variables must be interpreted as at the time of announcement rather than the actual value at transaction close. As variables (especially transaction composition and ownership structure) may change drastically between announcement and close, readers may also be biased to misinterpret the impact of these variables. The primary impact of these assumptions is that more narrow conclusions must be drawn from the findings.

## 7. Conclusions

There is widespread potential for the adoption of intelligent accounting methods across the entire financial industry, from automating data pipelines to algorithmic trading. This study highlights one such use—the identification of company features relevant to the market performance of Special Purpose Acquisition Companies. The primary contribution of the effort is in identifying target firm characteristics as potential determinants, in addition to SPAC, transaction, and market characteristics suggested in the prior literature. The key takeaway from this analysis is that SPAC sponsors should evaluate prior corporate/private equity backing within potential target firms, as the logistic regression model results indicate that prior corporate/private equity backing can increase the odds of SPAC market outperformance at 12 months post-merger by as much as $16\times$. These results are further corroborated by the linear regression model with fixed effects, which suggests these types of firms return an average of +10% after one year, even when accounting for time and industry fixed effects.

As to the reliability of these findings, the out-of-sample accuracy results, particularly for the purist out-of-sample data tranche tested at the conclusion of this study, indicate that the model can reliably classify both market under and overperformers at an accuracy rate of 67%. Evaluating these results in context, approximately 75% of the sample were underperformers and 25% were overperformers. As compared to choosing proportionally at random, the model is slightly worse at identifying underperformers, but almost 2.5 times better at identifying overperformers. This suggests that while corporate/private equity backing may be indicative of market overperformance, the lack of corporate/private equity backing is not necessarily indicative of an underperformance.

Additional variables suggested by this analysis to be influential, but not conclusively proven to be deterministic, include the announced enterprise value, early/late announcement, venture backing, and ownership structure. A summary of all the key findings from this research effort is provided below:

- Approximately 25% of SPACs outperform their exchange during the first year, generating a median return of +74%, with a median return range of +44% to +148%.
- Approximately 75% of SPACs underperform their exchange during the first year, generating a median return of −45%, with a median return range of −15% to −66%.
- Prior corporate/private equity involvement in target companies result in consistently increased odds of market outperformance and was demonstrated to have a statistically significant impact on SPAC returns at 12 months post-close.
- Announced enterprise value, early/late announcement, venture backing, and ownership structure all show at least some influence on market under-/overperformance but do not appear as statistically significant predictors across multiple models.

This study additionally creates implications for practitioners, notably, that a viable sponsor strategy could be developed around "following the money" and selecting target firms with prior corporate or private equity involvement. The basic premise for such a strategy would be to leverage corporate/private equity investors as credibility signals for identifying private-sector firms that are more capable of performing on the public market. Beyond the context of SPACs, the applications of this research are also intended to be broadly extensible to financial feature engineering efforts. In a machine learning context, this study demonstrates a robust process for moving from a wide array of potentially influential input variables to a small set of critically important feature predictors for market

securities. As demonstrated here, these feature predictors can be used for production models—achieving far greater accuracy than a simple weighted average model—as well as form the basis of traditional hypothesis-driven testing for foundational accounting research.

**Funding:** This research received no external funding.

**Data Availability Statement:** Restrictions apply to the availability of these data. Certain data were obtained from non-publicly available sources (e.g., Pitchbook) and is only available upon request from the corresponding author with permission from the respective third parties.

**Acknowledgments:** Special thank you to Eugene Fama and Nicholas Polson of The University of Chicago Booth School of Business for their inspiration, advice, and critiques throughout various stages of this research effort, and to Dulany Wagner, who shaped much of my initial exploration on the topic. Additionally, thank you to the many members of my Research Projects in Finance cohorts who took the time to share their thoughts, ideas, and feedback over the past two years.

**Conflicts of Interest:** This research has no conflicts of interest to report.

## Appendix A

**Table A1.** Venture involvement cluster variables and membership.

| Cluster | (1) | (2) | (3) | (4) |
|---|---|---|---|---|
| Venture-backed | 1 | 1 | 1 | 0 |
| Venture Funding > USD 300 M | 0 | 1 | 0 | 0 |
| Concurrent Round | 0 | 1 | 1 | 0 |
| Valuation Step-up > 10× [1] | 1 | 0 | 0 | 0 |
| Observations | 75 | 52 | 45 | 138 |

[1] Since last known valuation in Pitchbook.

**Table A2.** Revenue projection cluster variables and membership.

| Cluster | (1) | (2) | (3) | (4) |
|---|---|---|---|---|
| Demonstrated Revenue | 1 | 0 | 1 | 1 |
| 1–3 Year Projection | 1 | 0 | 0 | 0 |
| 4–5 Year Projection | 0 | 0 | 0 | 1 |
| 5+ Year Projection | 0 | 0 | 1 | 0 |
| CAGR > 30% | 0 | 0 | 1 | 1 |
| Observations | 116 | 62 | 54 | 78 |

**Table A3.** Transaction sources and uses cluster variables and membership.

| Cluster | (1) | (2) | (3) | (4) |
|---|---|---|---|---|
| Cash in Trust | - | 18% | 61% | 40% |
| Cash on B/S | - | 1% | 1% | 1% |
| PIPE Investment | - | 10% | 32% | 19% |
| Roll-over Equity | - | 68% | 3% | 15% |
| New/Existing Debt | - | 2% | 0% | 22% |
| Sponsor Shares | - | 1% | 0% | 1% |
| Other Sources | - | 0% | 2% | 2% |
| Roll-over Equity | - | 68% | 3% | 16% |
| Cash to B/S | - | 21% | 70% | 8% |
| Cash Consideration | - | 5% | 13% | 33% |
| Debt Repayment | - | 2% | 3% | 34% |
| Fees and Expenses | - | 2% | 10% | 5% |
| Sponsor Shares | - | 1% | 0% | 1% |
| Other Uses | - | 1% | 1% | 3% |
| Cash in Trust | - | 18% | 61% | 40% |
| Observations | 23 | 213 | 38 | 36 |

**Table A4.** Ownership cluster variables and membership.

| Cluster | (1) | (2) | (3) | (4) |
|---|---|---|---|---|
| Existing Shareholders | - | 74% | 24% | 56% |
| Public Shareholders | - | 13% | 27% | 31% |
| PIPE Shareholders | - | 10% | 23% | 5% |
| SPAC Sponsors | - | 3% | 5% | 7% |
| Other | - | 1% | 2% | 1% |
| Observations | 45 | 161 | 39 | 65 |

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
