# Peer review of "Picking Winners: Identifying Features of High-Performing Special Purpose Acquisition Companies (SPACs) with Machine Learning"

_jrfm, doi:10.3390/jrfm16040236_

Round 1

Reviewer 2 Report

The research utilizes machine learning models to identify influential market under/overperformance features within a sample of 310 SPACs closing acquisitions between 2005 and 2022. This approach is interesting, but I believe it should be improved by considering the points below.

1. Why did the authors not consider using the under/oversampling methods, such as SMOTE, to overcome the imbalanced property of the dataset?

2. The accuracy is not a suitable performance metric for the imbalanced dataset. How about using other metrics, such as the F1 scores, instead of that?

3. It is well-known that ensemble-based methods, such as random forest and gradient boosting, may be best suited for structured data. The machine learning methods also provide proper values to distinguish important features. Therefore, the authors should consider applying these methods also.

Lastly, there are many typos. See lines 143, 144, 220, etc.

Reviewer 3 Report

The goal of the paper is to identify Features of High-performing Special Purpose Acquisition Companies (SPACs) with Machine Learning. Authors provide an extensive overview of SPACs companies, and apply basic machine learning algorithms to identify the targeted companies. The paper is overall too simplistic, and I do not recommend it for the publication unless the serious rewritting is conducted. I would suggest authors to check the best practice in presenting similar reserach in the JRFM journal. One good example is: 

(2021). Churn management in telecommunications: Hybrid approach using cluster analysis and decision trees. Journal of Risk and Financial Management14(11), 544.

1. Abstract is not written in an usual manner. The abstract should be rewritten. Please, form the abstract in the following manner. First, describe the background of the research (1-2 sentences). Second, describe the research goals (1-2 sentences). Third, describe briefly (1-2 sentences) the methodology used. Fourth, describe the research results and conclusion in 3-4 sentences.

2. Literature reivew is too short. There should be an extensive review of the previously used modelling of the prediction of companies performance, with the focus to SPACs. Even if there is no previouis research in this area, the other applications for similar topic should be overviewed. Suggested reference is: (2019). Shedding light on the Doing Business Index: A machine learning approach. Business Systems Research: International journal of the Society for Advancing Innovation and Research in Economy10(2), 73-84.

3. The issue of using intelligent methods in accounting should be addressed. here is one possible reference:

(2019). Digitalization and the Challenges for the Accounting Profession. ENTRENOVA-ENTerprise REsearch InNOVAtion5(1), 428-437.

4. The issue of inbalanced dataset should be addressed. Here is one suggested reference for this issue:

(2022). Customer Response Model in Direct Marketing: Solving the Problem of Unbalanced Dataset with a Balanced Support Vector Machine. Journal of Theoretical and Applied Electronic Commerce Research17(3), 1003-1018.

5. The paper is not written in an usual style in terms of methodology. First, authors should elaborate the dataset used, then methodology and finally the algorithms planned to be used.

6. Result section is also scattered, it should be more condensed. 

7. Results of the regression modelling are in the conclusion, which should be removed tot the discussion section. 

8. In the last section, please focus on “Discussion, Implication, and Conclusion” to include (1).     Summary of the research - what was the goal, and how was it attained (2).    Theoretical implications - Discuss why the authors found these results and how they comply (or do not) with the Literature Review. (3).     Managerial Implications - Discuss why and how your results are relevant to the practice. (4).     Limitations of the paper (5).     Future Studies and Recommendations  

Reviewer 4 Report

The idea of the paper is very interesting and can contribute to industry and academia. However, the way the paper has been prepared is not very promising. 

The presentation of the figure and equation is very weak. 

Technical issue: The regression results are not good and need to be revised. 

Some other approaches should be used. The results in Table 7 confirm my observations. 

Round 2

Reviewer 1 Report

Accept in present form

Reviewer 2 Report

The authors have fully considered and addressed my concerns. I recommend that this study be published in a journal.

Reviewer 3 Report

Dear authors, the paper is now acceptable for the publication. 

Please, change the following reference:

Gulin, Danimir, Mirjana Hladika, and Ivana Valenta. “Digitalization and the Challenges for the Accounting Profession.” SSRN Scholarly 714 Paper. Rochester, NY, September 12, 2019. https://doi.org/10.2139/ssrn.3492237.

with the correct one:

Gulin, D., Hladika, M., & Valenta, I. (2019). Digitalization and the Challenges for the Accounting Profession. ENTRENOVA-ENTerprise REsearch InNOVAtion5(1), 428-437.

Reviewer 4 Report

The author has improved the quality of the paper, but still, grammar checks and language editing are needed. I suggest improving the language.